# Ileocolic Intussusception Due to Low-Grade Appendiceal Mucinous Neoplasm

**DOI:** 10.3390/diagnostics14182040

**Published:** 2024-09-14

**Authors:** Zhao-Yu Hsieh, Keng-Li Lin

**Affiliations:** 1Division of Medical Imaging, Far Eastern Memorial Hospital, New Taipei City 22060, Taiwan; 2Division of Colorectal Surgery, Far Eastern Memorial Hospital, New Taipei City 22060, Taiwan; calsdark.femh@gmail.com

**Keywords:** ileocolic intussusception, low-grade appendiceal mucinous neoplasm, appendiceal neoplasm, appendiceal mucocele, adult intussusception

## Abstract

We report a rare case of low-grade appendiceal mucinous neoplasm (LAMN) causing ileocolic intussusception. The case underscores the importance of considering ileocolic intussusception in differential diagnoses for nonspecific gastrointestinal symptoms. Early diagnosis via contrast-enhanced CT and scrupulous surgical intervention are crucial for favorable outcomes.

A 66-year-old woman with type 2 diabetes mellitus and hypertension presented to the emergency department with intermittent epigastralgia for several days. She denied fever, nausea, vomiting, diarrhea, constipation, black or bloody stool, and back pain. Her prior surgical, social, trauma, and family history were unremarkable. Physical examination showed epigastric tenderness, no obvious palpable mass, and no peritoneal sign. Laboratory data revealed no leukocytosis (9890/μL), no elevation of C-reactive protein (0.444 mg/dL), and mild anemia (hemoglobin: 10.8 g/dL). A bedside echo showed a suspected mass in the central abdomen. Contrast-enhanced abdominal CT revealed invagination of the terminal ileum, cecum, and mesentery into the transverse colon, causing mechanical ileus (Figure 1). A tubular, rim-enhancing cystic lesion with mural calcification was inside the intussusception. Ileocolic intussusception due to an appendiceal mucocele was considered. There was no decreased bowel mural enhancement, no strangulation, and no perforation. No ascites and no peritoneal deposits were present. Exploratory laparotomy showed a 3 cm sized swelling protruding lesion at the appendiceal base inducing intussusception until distal transverse colon (Figure 2). No evidence of peritoneal implants was present. Slow manual reduction of intussusception and an emergent right hemicolectomy with side-to-side anastomosis were performed (Figure 3).

Pathology revealed an inverted appendix with a dilated lumen filled with acellular mucin, accompanied by flattened and undulating columnar epithelium with low-grade mucinous cytoplasm (Figure 4). The mucin dissected into the subserosa and involved the adjacent cecum. High-grade cytologic atypia and infiltrative growth were not identified. Ileocecal intussusception due to low-grade appendiceal mucinous neoplasm (LAMN), classified as pT3 disease, was diagnosed.

Intussusceptions mainly occur in children and are usually attributed to lymphoid hyperplasia within the bowel wall. They are rare in adult populations and mostly have identifiable pathological lead points [1,2]. Symptoms vary from intermittent cramping, constipation to mechanical ileus. Serious complications may include bowel necrosis, sepsis, peritonitis, and bowel perforation. Ileocecal intussusception due to a LAMN is an extremely rare presentation, with very few cases published in the literature.

An appendiceal mucocele is a rare disease characterized by an enlarged and mucus-filled appendix. The causes include chronic obstruction of the appendix and neoplasms. Neoplastic mucoceles are classified as mucinous adenoma, LAMN, and appendiceal adenocarcinoma by the 2019 World Health Organization. LAMNs have an incidence of less than 0.3% of all appendectomy specimens [3]. They are more common in adult women in the fifth and sixth decades of life. Around 25% of mucoceles are asymptomatic and found incidentally, while they could cause right lower abdominal pain similarly to acute appendicitis [4,5]. Other clinical manifestations include an asymptomatic palpable mass, gastrointestinal bleeding, and increased abdominal circumference. Various clinical symptoms and a low incidence rate are related to misdiagnosis, delayed diagnosis, and a lack of prompt surgical management, which could lead to rupture of malignant mucoceles into the peritoneum causing pseudomyxoma peritonei [4,6].

Preoperative diagnosis is crucial for surgery planning. Preoperative contrast-enhanced CT has a diagnostic accuracy of 89.7% for appendiceal mucinous neoplasm [7]. CT scanning is also sensitive to identify the intussusception and characterize the lead point. Curvilinear mural calcification is highly suggestive of mucoceles but is present in less than 50% of cases. Simple mucoceles and mucinous adenomas are both encapsulated, low-attenuation cysts and indistinguishable. LAMNs demonstrate contrast-enhanced mural nodules, which could not be depicted in this case. Large irregular masses with thickened nodular walls are present in mucinous adenocarcinomas. Pseudomyxoma peritonei is characterized by loculated ascites and scalloped surfaces of the liver and the spleen [8].

A comprehensive literature search was conducted on appendiceal LAMNs with intussusception in the PubMed and Google Scholar databases. Relevant articles published in the last 10 years were collected and thoroughly reviewed. A total of 15 case reports were found (Appendix A) [5,9,10,11,12,13,14,15,16,17,18,19,20,21,22]. Twelve of the patients were female, and the other three were male. Their ages ranged from 30 to 80 years. Symptoms included intermittent or progressive abdominal pain, accompanied by nausea, vomiting, or other related manifestations. Mucoceles at the lead point were found in 12 cases, with three cases showing mural calcification and one case exhibiting inner hydro-aerial level. Enhancing mural nodules inside the mucoceles could not be found in these cases. The sizes of the appendiceal mucoceles ranged from 3.5 cm to 10.5 cm, with an average size of 6.24 cm. A total of 10 cases were classified as carcinoma in situ, 1 case had pT3 disease, and the other 4 cases did not report pathological staging. LAMNs are localized, low-grade neoplasms and have a favorable prognosis after resection. However, they may penetrate the appendicular wall, let mucinous deposits spread into the peritoneal cavity, and cause pseudomyxoma peritonei. A simple appendectomy is suggested for a local disease. A laparoscopic right hemicolectomy may be needed for a clear surgical margin if there are either cellular or acellular mucin on the serosal surface of the appendix or the mesoappendix or there is positive regional lymphadenopathy [23]. The operating surgeons performed a careful excision of the appendiceal mass in this patient, and no evidence of recurrent disease was present at the 40-month follow-up.

## Figures and Tables

**Figure 1 diagnostics-14-02040-f001:**
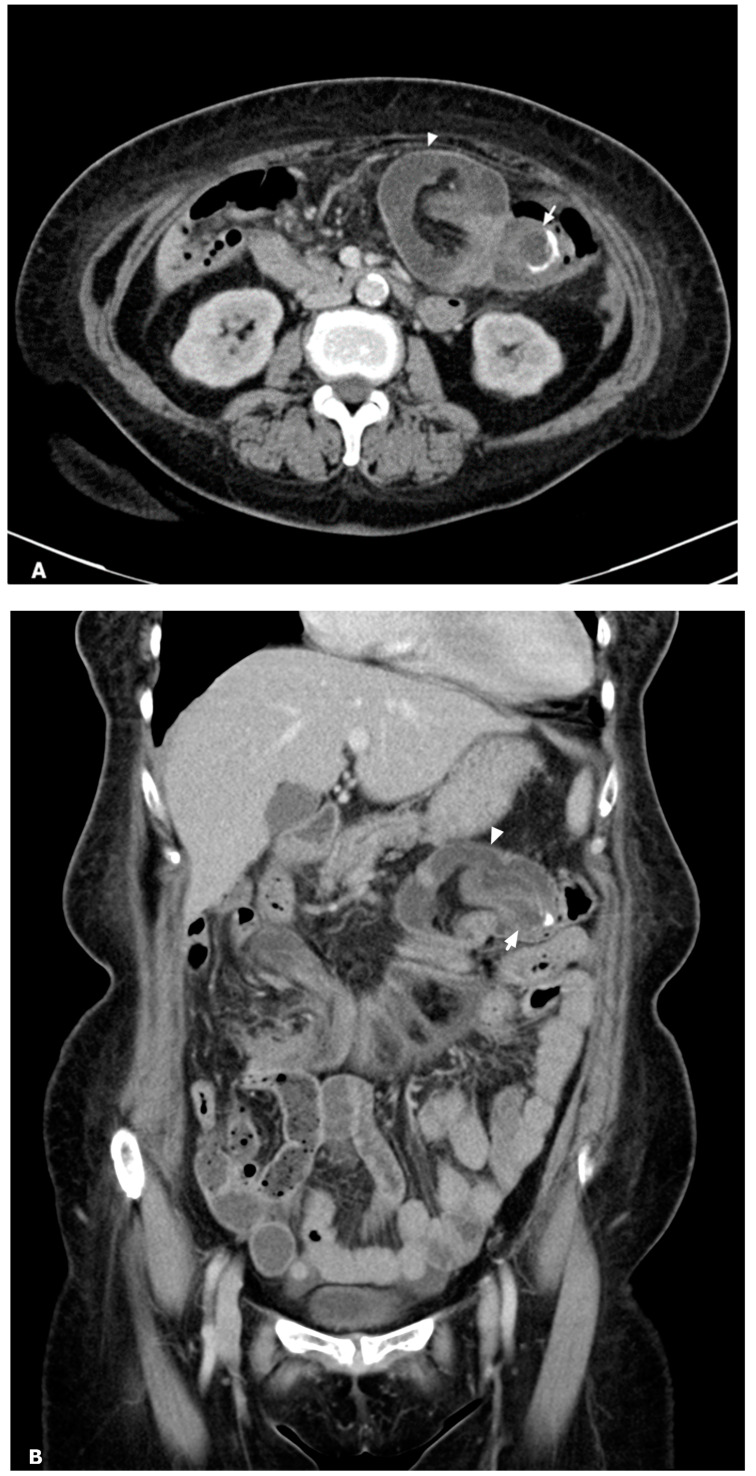
Contrast-enhanced CT axial (**A**) and coronal (**B**) reformatted images: a cystic dilated appendiceal base with mural calcification (arrow) as a lead point causing invagination of the terminal ileum, cecum, and mesentery into the transverse colon (arrowhead). An appendiceal mucocele leading to ileocecal intussusception was found.

**Figure 2 diagnostics-14-02040-f002:**
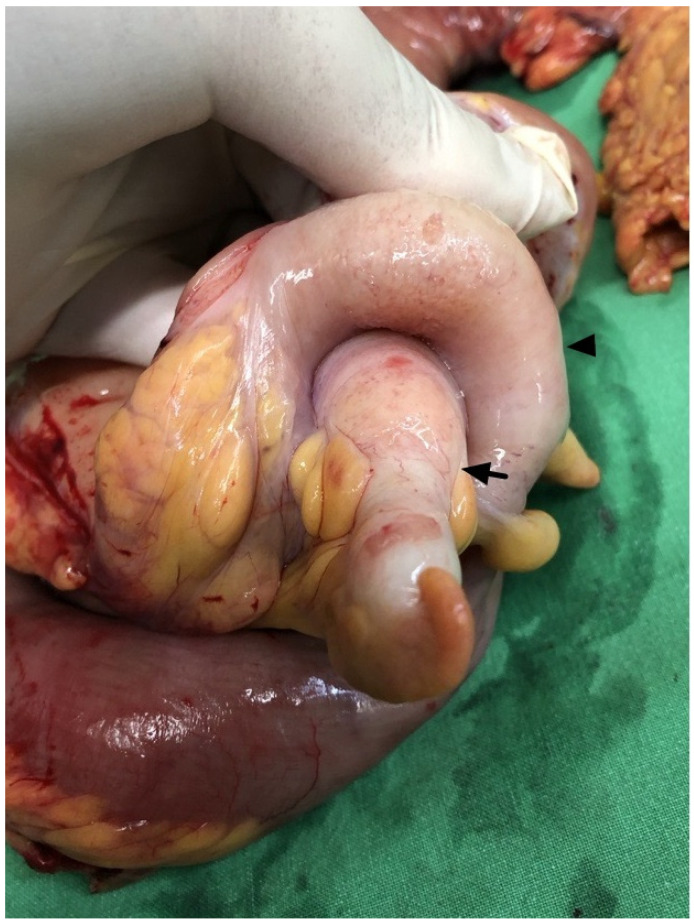
Operative findings: a focal protruding lesion at the inverted appendiceal base (arrow) invaginated into the cecum (arrowhead). No bowel cyanosis was present.

**Figure 3 diagnostics-14-02040-f003:**
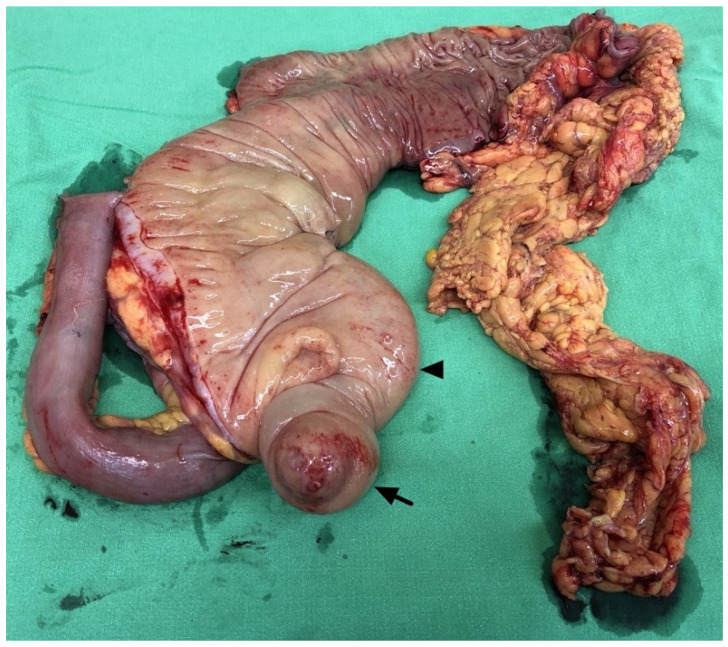
Gross specimen: right hemicolectomy specimen demonstrating the swollen appendix (arrow) and the edematous cecum (arrowhead). Ileocecal intussusception had been reduced.

**Figure 4 diagnostics-14-02040-f004:**
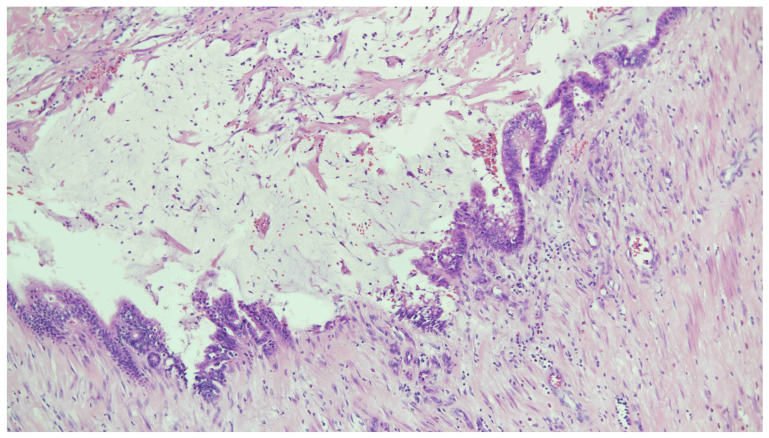
Histopathology with H&E staining: appendiceal tumor with abundant dissecting mucin pools and residual lining epithelium resting on the subjacent fibrous stroma. Columnar epithelium with low-grade atypia and variable apical mucinous cytoplasm was demonstrated.

## Data Availability

All relevant data are within the manuscript.

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
