# Peer review of "Ileocolic Intussusception Due to Low-Grade Appendiceal Mucinous Neoplasm"

_diagnostics, 2024, doi:10.3390/diagnostics14182040_

Round 1

Reviewer 1 Report

Comments and Suggestions for Authors

1. The patient's CRP value should be added.

2. The absence of a pathological specimen image is a vital deficiency and should be added.

Author Response

Comments 1: The patient's CRP value should be added.

Response 1: Thank you for pointing this out. We have added the CRP value in the manuscript at paragraph 1, page 1, line 38-39.

Comments 2: The absence of a pathological specimen image is a vital deficiency and should be added.

Response 2: We agree with this comment. A pathological specimen image has been added as “Figure 4”.

Reviewer 2 Report

Comments and Suggestions for Authors

An interesting case on ileocolic intussusception due to appendiceal mucinous neoplasm. The quality of the radiological and operative images is good and the presentation of the case is detailed and very nicely written, so kudos to the authors. Since the pathology is rare, I think this case could be of interest also for residents and practitioners of radiology and surgery. I have no further comments on this case.

Author Response

Comments 1: An interesting case on ileocolic intussusception due to appendiceal mucinous neoplasm. The quality of the radiological and operative images is good and the presentation of the case is detailed and very nicely written, so kudos to the authors. Since the pathology is rare, I think this case could be of interest also for residents and practitioners of radiology and surgery. I have no further comments on this case.

Response 1: Thank you a lot for your positive feedback and kind words regarding our case report on ileocolic intussusception due to an appendiceal mucinous neoplasm. We are grateful for your time and comments.

Reviewer 3 Report

Comments and Suggestions for Authors

This case is rare and very interesting, but it contains small problems. Thus, it is not acceptable for publication in the present form..

Minor points

1.     English: To be revised.

2.     Please enrich the Discussion part, by adding the general tendency of this rare association. For it, a brief but clear review of similar cases is necessary.

3.     Please discuss briefly the diagnostic differentiation between mucinous adenoma, LAMN, and mucinous adenocarcinoma by CT.

Author Response

Comments 1: English: To be revised.

Response 1: Thank you for pointing this out. We have reviewed the grammar and made the necessary modifications accordingly. 

Comments 2: Please enrich the Discussion part, by adding the general tendency of this rare association. For it, a brief but clear review of similar cases is necessary.

Response 2: We agree with this comment. We have expanded the Discussion section by including a brief but comprehensive review of similar cases and the imaging findings at paragraph 6, page 2, line 22-30 and “Table S1”.

Comments 3: Please discuss briefly the diagnostic differentiation between mucinous adenoma, LAMN, and mucinous adenocarcinoma by CT.

Response 3: Agree. We had a concise discussion on the diagnostic differentiation between mucinous adenoma, LAMN, and mucinous adenocarcinoma using CT imaging at the paragraph 5, page 2, line 12-21.

We hope these revisions meet your expectations and enhance the overall quality of our manuscript. Thank you again for your constructive comments.

Round 2

Reviewer 1 Report

Comments and Suggestions for Authors

The necessary changes have been made, and I have no additional suggestions.